# Expression of CD47 and SIRPα Macrophage Immune-Checkpoint Pathway in Non-Small-Cell Lung Cancer

**DOI:** 10.3390/cancers14071801

**Published:** 2022-04-01

**Authors:** Alexandra Giatromanolaki, Achilleas Mitrakas, Ioannis Anestopoulos, Andreas Kontosis, Ioannis M. Koukourakis, Aglaia Pappa, Mihalis I. Panayiotidis, Michael I. Koukourakis

**Affiliations:** 1Department of Pathology, Medical School, Democritus University of Thrace, 68100 Alexandroupolis, Greece; agiatrom@med.duth.gr (A.G.); amitrak@med.duth.gr (A.M.); ioannisa@cing.ac.cy (I.A.); 2Department of Cancer Genetics, Therapeutics & Ultrastructural Pathology, The Cyprus Institute of Neurology & Genetics, Nicosia 2371, Cyprus; mihalisp@cing.ac.cy; 3The Cyprus School of Molecular Medicine, The Cyprus Institute of Neurology & Genetics, Nicosia 2371, Cyprus; 4Department of Molecular Biology & Genetics, Democritus University of Thrace, 68100 Alexandroupolis, Greece; andreas.kontosis@luhs.org (A.K.); apappa@mbg.duth.gr (A.P.); 5Department of Radiotherapy/Oncology, Medical School, Democritus University of Thrace, 68100 Alexandroupolis, Greece; koukourioannis@gmail.com

**Keywords:** lung cancer, CD47, SIRPα, CD68, PD-L1, FOXP3, prognosis

## Abstract

**Simple Summary:**

Cancer cells escape macrophage phagocytosis by exploiting the CD47/SIRPα axis. We found that extensive membranous CD47 expression by cancer cells characterized 29/98 cases. SIRPα and CD68 were expressed by tumor-associated macrophages (Μφ, TAMs). A high CD68Mφ-score was linked with improved overall survival. High expression, however, of SIRPα by CD68+ TAMs was linked with CD47 expression by cancer cells, low TIL-score, and poor prognosis. A direct association of CD47 expression by cancer cells and the % FOXP3+ TILs was also noted. The CD47/SIRPα axis is a sound target for adjuvant immunotherapy policies, aiming to improve the cure rates in operable NSCLC.

**Abstract:**

Background: Cancer cells escape macrophage phagocytosis by expressing the CD47 integrin-associated protein that binds to the SIRPα ligand (signal regulatory protein alpha) expressed by macrophages. Immunotherapy targeting this pathway is under clinical development. Methods: We investigated the expression of CD47/SIRPα molecules in a series of 98 NSCLCs, in parallel with the infiltration of tumor stroma by CD68+ macrophages, tumor-infiltrating lymphocytes (TILs), and PD-L1/PD-1 molecules. Results: Extensive membranous CD47 expression by cancer cells characterized 29/98 cases. SIRPα and CD68 were expressed, to a varying extent, by tumor-associated macrophages (Μφ, TAMs). A high CD68Mφ-score in inner tumor areas was linked with improved overall survival (*p* = 0.005); and this was independent of the stage (*p* = 0.02, hazard ratio 0.4). In contrast, high SIRPα expression by CD68+ TAMs (SIRPα/CD68-ratio) was linked with CD47 expression by cancer cells, low TIL-score, and poor prognosis (*p* = 0.02). A direct association of CD47 expression by cancer cells and the % FOXP3+ TILs (*p* = 0.01, r = 0.25) was also noted. Conclusions: TAMs play an important role in the prognosis of operable NSCLC. As SIRPα+ macrophages adversely affect prognosis, it is suggested that the CD47/SIRPα axis is a sound target for adjuvant immunotherapy policies, aiming to improve the cure rates in operable NSCLC.

## 1. Introduction

The development of monoclonal antibodies targeting immune-checkpoint pathways opened the most promising current era of cancer immunotherapy. A large number of therapeutic antibodies have been approved for clinical applications, which at present target the PD-L1/PD-1 pathway and the CTLA4 inhibitory ligand of CD80/CD86 molecules. Both pathways aim to enhance the cytotoxic activity of T-cells [1]. Another important immune-checkpoint pathway is the CD47/SIRPα one. Normal cells escape the phagocytic activity of macrophages by expressing the CD47 (integrin-associated protein, IAP) glycoprotein that binds to the SIRPα ligand (signal regulatory protein alpha) expressed by macrophages [2]. Cancer cells can exploit the same pathway by overexpressing CD47 to mask their alien nature against macrophages. Consequently, anti-CD47 and anti-SIRPα antibodies are under preclinical and clinical evaluation [3,4].

The expression patterns of CD47 in human malignancies have been investigated in a small number of published studies, mainly from Eastern Institutes. Association with aggressive clinical behavior and poor prognosis have been reported in breast [5], esophageal [6] and gastric cancer [7], as well as hepatomas [8], and melanomas [9]. In non-small cell lung cancer (NSCLC), contradictory results have been published in four reports [10,11,12,13]. Regarding SIRPα expression, its clinicopathological and prognostic value in solid tumors is under investigation. Contradictory results have been reported in lymphomas [14,15]. A recent study in esophageal cancer showed that SIRPα was directly linked with invasive disease and poor prognosis [16]. In colorectal cancer high CD47 expression was linked with poor prognosis, whilst the high presence of SIRPα macrophages was linked with a favorable outcome [17].

In the current study, we investigated the CD47/SIRPα axis in a series of NSCLCs by examining in parallel the immunohistochemical patterns of expression of CD47, SIRPα, and CD68 pan-macrophage marker. Their correlation with histopathological characteristics and the postoperative prognosis is reported. Moreover, we analyze the association of CD47/SIRPα with the PD-L1/PD-1 axis and with parameters related to tumor-infiltrating lymphocytes.

## 2. Results

### 2.1. Expression in Normal Lungs

In a normal lung, CD47 was expressed by the bronchial epithelium (Figure 1a), alveoli (Figure 1b), vessels, and seromucinal glands, while chondrocytes of the bronchial cartilage were negative. Alveolar macrophages were also positive. CD68 and SIRPα were not expressed in normal lung tissues. SIRPα and CD68 stained alveolar macrophages (Figure 3a,b).

### 2.2. CD47 Expression in Cancer

CD47 was expressed in the membrane and cytoplasm of cancer cells (Figure 1c,d). The percentage of cancer cells with membrane CD47 reactivity ranged from 0–90% (median 20%). Tumors were grouped into three categories according to the percentage of positive cancer cells: negative (0% or sporadic positive cells; 33 cases), medium (10–40% positive cancer cells; 36 cases), and high (50–90% positive cancer cells; 29 cases). Analysis, according to stage showed a significant association of high CD47 expression with stage 3 disease (13/29 vs. 17/69; *p* = 0.05). There was no association with histology type (*p* = 0.72). Out of 58 squamous cell carcinomas 21 (36.2%) were negative for CD47, 20 (34.5%) had medium expression and 17 (29.3%) had high expression. Out of 22 adenocarcinomas, 7 (31.8%) were negative, 7 (31.8%) had medium expression and 8 (36.4%) had high expression. Out of 18 undifferentiated tumors, 5 (27;8%) were negative, 9 (50%) had medium expression and 4 (22.2%) had high expression.

Similarly, there was no association of CD47 expression with and MIB1 proliferation index (*p* = 0.83). Out of 33 tumors with a low proliferation index 11 (33.3%) were negative for CD47, 10 (30.3%) had medium expression and 12 (36.4%) had high expression. Out of 46 tumors with a medium proliferation index, 15 (32.6%) were negative, 19 (41.3%) had medium expression and 12 (26.1%) had high expression. Out of 19 tumors with a high proliferation index, 7 (36.8%) were negative, 7 (36.8%) had medium expression and 5 (26.4%) had high expression.

Kaplan–Meir analysis of disease-specific survival did not reveal any significant association with CD47 expression when considering all cases or stratified for stage (Figure 2a).

CD47 was expressed by stromal fibroblasts, in 19/98 cases (Figure 1e). There was no association with CD47 expression by cancer cells, histopathological parameters, MIB1 proliferation index, or patients’ prognosis.

### 2.3. SIRPα and CD68 Expression in Cancer

SIRPα and CD68 were not expressed by cancer cells but rather by tumor-associated macrophages (Μφ, TAMs) (Figure 3). The extent of expression varied among cases, ranging from sporadic expression of Mφ to an expression involving the entire Mφ population. The number of SIRPα+ and CD68 + Μφ per ×40 optical field was quantified, and cases were categorized into four groups (score 0,1,2,3), as indicated in the ‘Materials and Methods’ section. The distribution of SIRPαΜφ-score and CD68+Mφ-score in the invading front and inner tumor areas is shown in Table 1. Figure 3 shows typical cases with a low, medium, and high density of Μφ presence in the tumor stroma, expressing CD68 or SIRPα. Appendix A shows typical images of confocal fluorescent microscopy after triple staining for nuclei (DAPI), CD68, and SIRPα, from two tumors with the extensive presence of CD68+/SIRPα+ and CD68+/SIRPα- macrophages, respectively. Linear regression analysis showed no significant association between the SIRPα and the CD68 scores in the invading tumor front (*p* = 0.17) or inner tumors areas (*p* = 0.90).

Linear regression analysis revealed a strong direct association of SIRPαΜφ-score between tumor invading front and inner tumor areas (*p* < 0.0001, r = 0.79). There was no statistical association between SIRPα-score, histology type, stage of disease, or MIB proliferation index. The mean SIRPαMφ-score in the invading front was 0.75, 0.95, and 0.72 in the squamous cell carcinomas, adenocarcinomas, and undifferentiated carcinomas, respectively (*p* > 0.17). The mean SIRPαMφ-score in inner tumor areas was 0.75, 0.81, and 0.66 in the squamous cell carcinomas, adenocarcinomas, and undifferentiated carcinomas, respectively (*p* > 0.38). The mean SIRPαMφ-score in the invading front was 0.78, 0.90 and 0.73 in stage 1, 2 and 3, respectively (*p* > 0.32). The mean SIRPαMφ-score in inner tumor areas was 0.69, 0.90 and 0.73 in stage 1, 2 and 3, respectively (*p* > 0.11). The mean SIRPαMφ-score in the invading front was 0.69, 0.76, and 0.1.05 in tumors with low, medium, and high MIB1 proliferation index, respectively. Although tumors with a high proliferation index had a higher SIRPαMφ-score, the statistical difference was marginal (*p* > 0.06). The mean SIRPαMφ-score in inner tumor areas was 0.69, 0.71, and 0.94 in tumors with low, medium, and high MIB1 proliferation index, respectively (*p* > 0.12).

In addition, the same analysis revealed a strong direct association of CD68Mφ-score between tumor invading front and inner tumor areas (*p* < 0.0001, r = 0.71). There was no association of the CD68Mφ-score with the histology, stage. The mean CD68Mφ-score in the invading front was 1.8, 2.1, and 2.2 in the squamous cell carcinomas, adenocarcinomas, and undifferentiated carcinomas, respectively (*p* > 0.15). The mean CD68Mφ-score in inner tumor areas was 1.8, 1.8, and 2.1 in the squamous cell carcinomas, adenocarcinomas, and undifferentiated carcinomas, respectively (*p* > 0.26). The mean CD68Mφ-score in the invading front was 2.1, 1.8 and 2.0 in stage 1, 2 and 3, respectively (*p* > 0.12). The mean CD68Mφ-score in inner tumor areas was 1.9, 1.7, and 1.9 in stages 1, 2 and 3, respectively (*p* > 0.27). MIB1 proliferation index was significantly higher in tumors with a high CD68Mφ-score, especially in inner tumor areas (11/35 vs. 8/63; *p* = 0.02).

### 2.4. SIRPα, CD68 and Survival

Kaplan–Meir analysis of disease-specific survival did not reveal any significant association with SIRPαΜφ-score (considering all cases), whether calculated in the invading tumor front or inner tumor areas (*p* > 0.95) (Figure 2b).

Furthermore, survival analysis of disease-specific survival (considering all cases) did not reveal any significant association with CD68Mφ-score when calculated in the invading tumor front (*p* = 0.31). A strong association of high CD68Mφ-score in inner tumor areas with better overall survival was noted (*p* = 0.005); Figure 2c. The impact of the CD68Mφ-score was more prominent in stage II/III tumors (stage I: *p* = 0.37; stage II/III: *p* = 0.05)

In a bivariate model of regression analysis of death events, a high CD68Mφ-score was an independent prognostic variable (*p* = 0.02, hazard ratio 0.4) together with the stage (*p* = 0.0001, Hazard ratio 3.93).

### 2.5. SIRPα/CD68 Ratio Analysis

The density of Mφ expressing SIRPα among CD68+ Μφ was scored by dividing the SIRPαMφ-score bythe CD68Mφ-score. This SIRPα/CD68-ratio ranged from 0 to 1, and its distribution is shown in Table 2. Despite the significant association between the SIRPα/CD68-ratio in the invading front and inner tumor areas (*p* < 0.0001, r = 0.75), the ratio differed in 30/98 cases.

A ratio of 0–0.3 was considered low, while 0.5–1 was high. Kaplan–Meir survival analysis of disease-specific survival (considering all cases) showed a direct association of high SIRPα/CD68-ratio in inner tumor areas with poor prognosis (*p* = 0.02; Figure 2d). There was no significant association with SIRPα/CD68-ratio when calculated in the invading tumor front (*p* = 0.74; Figure 2e).

### 2.6. Association of CD47 with SIRPα and CD68 Scores

Linear regression analysis revealed a direct association between CD47 expression by cancer cells and SIRPαΜφ-score (*p* = 0.0008, r = 0.33 for both the invading front and inner tumor areas; Figure 4a,b). Out of 33 cases with negative CD47 expression 19 (57.5%) did not have SIRPα+ Mφs. In contrast, out of 65 cases with medium/high expression of CD47, only 11 (16.9%) showed a lack of SIRPα Mφs (*p* < 0.0001). Thus, 54/98 NSCLCs expressing CD47 also contained SIRPα+ infiltrating Mφs. Out of 29 NSCLCs with a high expression of CD47, 25 were also infiltrated by SIRPα expressing Μφs. A direct association was also noted with the SIRPα/CD68-ratio (*p* = 0.04, r = 0.21 for the invading front; *p* = 0.03, r = 0.21 for inner tumor areas; Figure 4c). There was no association with CD68Μφ-scores. Overall survival analysis stratifying for CD47/SIRPα did not reveal any subgroups of patients with a difference in prognosis.

### 2.7. Correlations with PD-L1/PD-1 and TILs

Analysis of CD47 expression and SIRPα and CD68 parameters did not show any association with PD-L1 expression by cancer cells or with the density of PD-1 positive TILs. A direct association of CD47 expression by cancer cells and the % FOXP3+ TILs (*p* = 0.01, r = 0.25) and with the FIL-score (*p* = 0.004, r = 0.28) was noted in linear regression analysis (Figure 4d,e). Moreover, the SIRPα/CD68-ratio was inversely related to the TIL-score (*p* = 0.001, Figure 4f). The SIRPα and CD68Mφ-score were not related to the TIL-score.

## 3. Discussion

Innate immunity exerted by macrophages, γδ-Τ-cells, and NK-cells play an important role in cancer cell recognition and destruction [18]. Direct cell killing is performed by secretion of granzymes and perforine, iNOS, or even TNFα. However, macrophages play a dual role in the tumor microenvironment, as their polarization towards an M1 or M2 phenotype drives cytotoxic or immunosuppressive responses, respectively. Production of VEGF and TGFβ by Μ2 macrophages block cytotoxic T-cell activity and promote tumor growth [19].

In the present study, we quantified the infiltration of invading tumor front and of inner tumor stroma by CD68 expressing macrophages. A dense presence of tumor-associated macrophages (TAMs) was recorded in one-third of tumors. Although no association was found with histopathological parameters, dense TAM infiltration was linked with a high MIB1 proliferation index. However, survival analysis showed that dense TAM presence was significantly related to a better postoperative outcome. This finding suggests that macrophage infiltration and related innate immunity does, indeed, exert anti-tumor activity in a fraction of tumors. Of interest, the survival benefit was more prominent in stage II/III disease. As for most patients with advanced local disease, death occurs from metastasis, our findings support the idea that recognition of tumors by macrophages may have contributed to the eradication of micro-metastatic disease till the time of the operation. Immunotherapeutic approaches that would enhance macrophage activity may, therefore, prove of value in the improvement of cure rates offered by surgery, in patients with compromised macrophage innate immunity.

Nonetheless, the CD68 marker does not discriminate between M1 or M2 macrophage phenotypes or immune checkpoint molecules exploited by cancer cells to block macrophage activity. In the current study, we focused on this latter issue, by investigating the CD47/SIRPα inhibitory checkpoint axis. The CD47/SIRPα axis is an important pathway in normal physiology that blocks the phagocytic activity of macrophages. SIRPα is a glycoprotein abundant in neurons and the myeloid lineage of hemopoietic cells, including macrophages. Expression of the CD47 integrin by most of the normal cells blocks the phagocytic activity of macrophages by binding to their extracellular domain of SIRPα. Mature red blood cells, for example, lose their CD47 molecules, becoming a target for their elimination from the bloodstream by macrophages [20]. Adopting a similar behavior, cancer cells overexpress CD47 to escape M1 macrophage innate immune response by binding to their SIRPα ligand. Increased expression of CD47 has been documented in hematological malignancies and also in various human solid tumors [5,6,7,8,9,10,11,12]. Restoring the innate macrophage immunity by targeting the CD47/SIRPα axis is under intense preclinical and clinical evaluation [3,4].

There are only a few studies published on the clinicopathological role of CD47 in NSCLC. Arrieta et al. examined tumor biopsies from 169 NSCLCs, showing expression of CD47 in 84% of tumors, and although there was no overall association with prognosis, CD47 defined better prognosis in patients with EGFR mutations [10]. Yang et al. also showed a frequent expression of CD47 in 68% of tumors examined, and this was co-expressed with PD-L1 in 18% of patients [12]. Xu et al. found a negative correlation between CD47 and PD-L1 expression in a series of NSCLCs and a positive correlation with EGFR mutations and poor prognosis [13]. Both CD47 expression and CD47/PD-L1 co-expression were liked with a poorer prognosis. Barrera et al. found that expression of CD47 by peripheral neutrophils was associated with resistance to phagocytosis and poor prognosis, but the authors did not report on tumor CD47 expression [11].

In our study, analysis of NSCLC patients showed that membrane CD47 overexpression by cancer cells was noted in two-thirds of tumor samples, where one-third expressed CD47 in more than 50% of cancer cells. This finding shows that, indeed, the majority of NSCLCs can mask themselves against the macrophages. In addition, CD47 overexpression by cancer cells was directly related to SIRPα expression by TAMs and not with CD68+ TAMs, showing that an interplay between CD47 expressing cancer cells and the SIRPα+ Μφ exists in NSCLC. This hypothesis was further substantiated in the survival analysis. Although dense CD68+ TAM presence was linked with good prognosis, expression of SIRPα by these TAMs (high SIRPα/CD68-ratio) was strongly linked with poor postoperative prognosis. Eventually, SIRPα emerges as a potent marker to identify immunosuppressive type M2 macrophages. As the CD47/SIRPα cancer cell/Mφ axis is commonly expressed in NSCLC a therapeutic blockage could prove useful as adjuvant or neo-adjuvant immunotherapy in operable NSCLC [21].

Analysis of TAMs and CD47/SIRPα expression patterns did not reveal any association with the expression of PD-L1/PD-1, another important immune checkpoint pathway related to cytotoxic T-cell activity in NSCLC. The direct association of CD47 expression, however, with poor stroma infiltration by TILs and, most importantly, with dense FOXP3+ TIL presence, is an additional interesting observation linking macrophage and T-cell immunosuppressive pathways in a subset of NSCLCs. This is in accordance with a previous study in gastric cancer, where CD47 expression was linked with a lower CD8/FOXP3 ratio [22]. Tseng et al. also found in in vivo experiments that treatment with anti-CD47 antibodies not only allows macrophage phagocytosis of cancer cells but also promotes an anti-tumor cytotoxic T-cell immune response and reduction of FOXP3+ regulatory T-cells [23]. It is, therefore, likely that immunotherapy targeting the CD47/SIRPα axis may enhance anti-tumor response by activating both macrophage-mediated innate and T-cell adaptive immunity.

## 4. Materials and Methods

### 4.1. Patient and Disease Characteristics

Ninety-eight sequential tissue samples of surgically resected NSCLC were retrieved from the archives of the Department of Pathology, Democritus University of Thrace. Tissue sections were selected to contain the invading tumor front of the tumor, so these also contained adjacent normal lung tissue. All patients were treated with surgery alone. The median age of patients was 68 years (range 32–82). According to the UICC staging system, 46/98 patients were of stage I, 22/98 of stage II, and 30/98 of stage III. Regarding histology, 58 cases were of the squamous type, 22 cases were adenocarcinomas, and 18 were undifferentiated carcinomas. Out of 58 squamous cell carcinomas, 30 were of grade 1, 15 of grade 2, and 13 of grade 3. Out of 22 adenocarcinomas, 7 were of grade 1, 10 of grade 2, and 5 of grade 3. The median follow-up of patients was 46 months, ranging from 26–112 months.

### 4.2. Ethical Considerations

The study was conducted according to the criteria set by the declaration of Helsinki, after approval from the Scientific and Ethics Committee of the University Hospital of Alexandroupolis (study approval number ES11-26-11-18).

### 4.3. Immunohistochemistry

Formalin-fixed paraffin-embedded tissue sections of 3 μm thick were cut and placed on poly-L-lysine slides. For CD47 detection, we used the primary mouse monoclonal ab260418 (abcam, Cambridge, UK), with 60 min incubation, at dilution 1/200. For the detection of SIRPα we used the primary rabbit monoclonal ab260039 antibody (abcam, Cambridge, UK), with 60 min incubation, at dilution 1/250. Immunohistochemistry for CD68 was performed using the mouse monoclonal KP1 clone (Immunologic) at dilution 1/300, with 60 min incubation. The heat-induced epitope retrieval process was performed in a microwave oven using Dako EnVision FLEX Target Retrieval Solution (pH 9.0). After washing the specimen twice for 6 min each time, the slides were incubated with the primary antibodies. Following washing with PBS, endogenous peroxidase was quenched with EnVision Flex Peroxidase Block (DAKO, Glostrup, Denmark) for 10 min. Non-specific binding was blocked in EnVision Flex mouse or rabbit Linker, as appropriate, for 15 min (DAKO) and washed in PBS. Subsequently, slides were incubated with the secondary antibody (EnVision Flex/HRP; DAKO) for 30 min. The color was developed by 5 min incubation with EnVision Flex Chromogen (DAKO), and sections were counterstained weakly with hematoxylin. In every staining run, the primary antibody was replaced by normal species-specific immunoglobulin-G from at least one section to provide a negative control.

Assessment of the expression of CD47 was performed at ×20 magnification. The percentage of cancer cells with membrane expression was recorded in all available optical fields (o.f.). The mean value was used to score each tissue section. The expression of CD47 by stroma fibroblasts was also quantified. The number of ×20 o.f. showing strong expression divided by the total number of ×20 o.f. was used to score the extent of CD47 expression in the stroma for each case.

The number of macrophages (Mφ) stained for SIRPα and CD68 was recorded in all ×40 magnification o.f., separately in the invading tumor front and the inner tumor stroma. As invading tumor front is defined the ×200 optical fields that contain the tumor are located in direct proximity to the normal lung tissue. All other optical fields, moving to central tumor areas, are considered inner tumor areas. The final grouping of cases according to the Μφ density (SIRPαΜφ-score and CD68Μφ-score) was performed as follows: Score 0: lack of expression, Score 1: less than 20 Mφ/o.f., Score 2: 20–50 Mφ/o.f. and Score 3: >50 Mφ/o.f. Subsequently, we calculated the SIRPα/CD68-ratio (for the invading front and inner tumor areas) by dividing the SIRPαΜφ-score/CD68Μφ-score. This ratio reflects the percentage of tumor-associated macrophages (TAMs) that express SIRPα.

### 4.4. Confocal Microscopy

A sequential immunofluorescence staining protocol in two steps was applied on cancer tissue sections from 5 patients with extensive CD68+ and SIRPα+ and 5 tissue sections with extensive CD68+ but low SIRPα+ macrophage stroma infiltration documented in immunohistochemistry single staining. The anti-CD68 and anti-SIRPα antibodies were used at concentrations and conditions used in immunohistochemistry. Details on the protocol applied have been previously published by our group [24]. The anti-rabbit Biotium-CF568 (1:250; Fremont, CA, USA) and the anti-mouse Biotium-CF488 (1:250; Fremont, CA, USA) secondary antibodies were used for 30 min incubation, for SIRPα and CD68 staining, respectively. The DNA was counterstained with Hoechst 33,342 for 30 min at room temperature. Image acquisition was performed on a customized Andor Revolution Spinning Disk Confocal system (Yokogawa CSU-X1) built around an Olympus IX81 with 10 × 0.30 NA air lens and a digital camera (Andor Ixon+885) (Bioimaging Facility, MBG-DUTH). The system was controlled by Andor IQ3 software. Images were acquired as z-stacks with a z-step of 2 μm, through the entire volume of the cells.

### 4.5. Scoring of TIL-Density

The quantification of tumor-infiltrating lymphocytes TIL-density in the tumor stroma has been previously described [25,26]. TILs were assessed in the CD47 immunostaining slides as lymphocytes stained with hematoxylin. This does not result in any bias because the lymphoid population is mainly stained with hematoxylin even after hematoxylin and eosin staining. The number of TILs was assessed in all ×40 optical fields and the mean value was obtained to score each case. Cases were subsequently grouped in four different TIL-scores as follows: 1 (minimal, mean value 1–10 lymphocytes/o.f.), 2 (low, mean value 10–70 lymphocytes/o.f.), 3 (medium, mean value 70–150 lymphocytes/o.f.), and 4 (high, mean value > 150 lymphocytes/o.f.).

### 4.6. Other Immunohistochemical Markers

We further assessed the expression of PD-L1 by cancer cells and of PD-1 and FOXP3 in tumor-infiltrating lymphocytes (TILs), using the rabbit monoclonal anti-PD-L1 antibody clone CAL10 (Biocare Medical, Pacheco, CA, USA), the mouse monoclonal clone NAT105 (Biocare Medical, Pacheco, CA, USA) and an ‘in house’ undiluted hybridoma supernatant from the well-validated anti-FOXP3 murine monoclonal antibody 236A/E7, respectively, as previously reported [26].

For PD-L1, the percentage of cancer cells with strong membrane (with or without cytoplasmic) expression was recorded in all ×200 o.f., and the mean value was calculated to score each case. Cases were grouped into three categories thus negative, low (expression in 1–9% of the cancer cells), and high (expression in >10% of the cancer cells).

The percentage of PD-1 and of FOXP3 expressing TILs among the total amount of eosin-stained TILs present in the tumor stroma was assessed in all ×40 o.f., and the mean score was calculated for each case. As this score provides the % of TILs expressing PD-1 or FOXP3, and does not reveal the extent of PD-1 and FOXP3 lymphocytic infiltration in the tissue, which also depends upon the extent of TIL presence. The *PIL-score* and the *FIL-score* were, therefore, assessed as the product of “TIL-score” ×“% PD-1 or FOXP3+ TILs”.

### 4.7. Statistical Analysis

For statistical analysis and graph presentation, we used the GraphPad Prism 7.0 software. The chi-square or Fisher’s exact t-test was used to compare categorical variables. The Pearson’s correlation analysis test was used to assess relations between continuous variables. The overall disease-specific survival was assessed with the Kaplan–Meier method. A Cox proportional hazard model was used for multivariate analysis of death events. A *p*-value of <0.05 was considered for significance.

## 5. Conclusions

In conclusion, TAMs play an important role in the control of metastasis and the prognosis of operable NSCLC. CD47/SIRPα up-regulation is a common event in NSCLCs, and SIRPα expression by macrophages is linked with poor prognosis. Whether CD47/SIRPα is a critical immune checkpoint target for the development of adjuvant immunotherapy policies, aiming to improve the cure rates in operable NSCLC should be investigated in clinical studies. Further studies on M1/M2 phenotypic prevalence in NSCLC are ongoing, anticipating a prognostic and therapy guiding classification of tumors according to the patient macrophage innate immunity status.

## Figures and Tables

**Figure 1 cancers-14-01801-f001:**
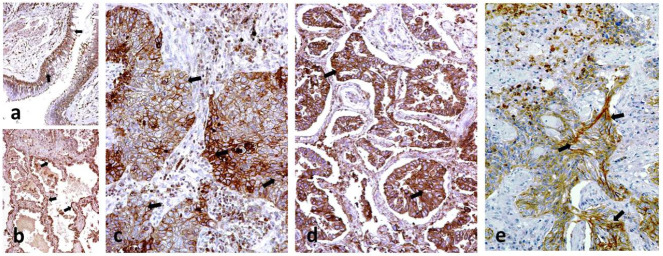
Immunhistochemistry for CD47: (**a**) CD47 immunostaining of normal bronchial epithelium (arrows); (**b**) CD47 immunostaining by alveolar epithelium (arrows); (**c**) Strong staining of CD47 expressed by the cellular membranes of cancer cells (arrows) in a squamous cell carcinoma of the lung; (**d**) Strong staining of CD47 expressed by the cellular membranes of cancer cells in lung adenocarcinoma (arrows); (**e**) CD47 expression by stroma fibroblasts (arrows) in the context of lack of CD47 expression by cancer cells; All images are shown at ×20 magnification.

**Figure 2 cancers-14-01801-f002:**
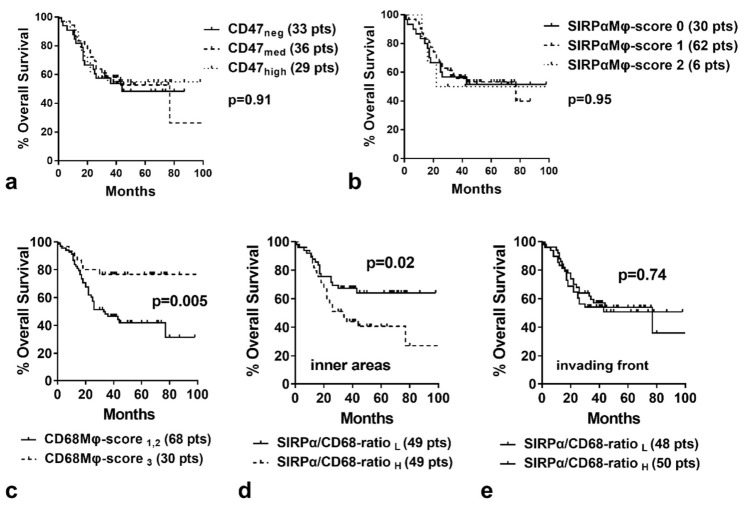
Kaplan–Meier disease-specific overall survival curves stratified for: (**a**) CD47 expression by cancer cells; (**b**) SIRPα-Μφ score; (**c**) CD68-Mφ score, (**d**) SIRPα/CD68-ratio in inner tumor areas, and (**e**) SIRPα/CD68-ratio in the invading tumor front (neg = negative, med = medium, pts = patients. L = low, H = high).

**Figure 3 cancers-14-01801-f003:**
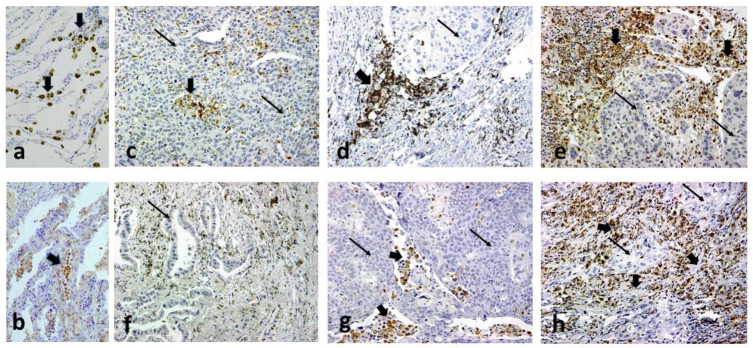
Immunohistochemical images of normal lung alveolar tissue showing presence of macrophages stained for CD68 (**a**) and SIRPAa (**b**). Typical immunohistohemical images of non-small cell lung carcinomas with (**c**) low infiltration of the tumor stroma by CD68+ macrophages (score 1); (**d**) medium infiltration of the tumor stroma by CD68+ macrophages (score 2); (**e**) intense infiltration of the tumor stroma by CD68+ macrophages (score 3); (**f**) lack of SIRPα+ macrophages in the tumor stroma (score 0); (**g**) low infiltration of the tumor stroma by SIRPα+ macrophages (score 1); (**h**) intense infiltration of the tumor stroma by SIRPα+ macrophages (score 2). All images are shown at ×20 magnification. Thick arrows show areas of macrophage presence. As noted in all images CD68 and SIRPα were not expressed by cancer cells (thin arrows).

**Figure 4 cancers-14-01801-f004:**
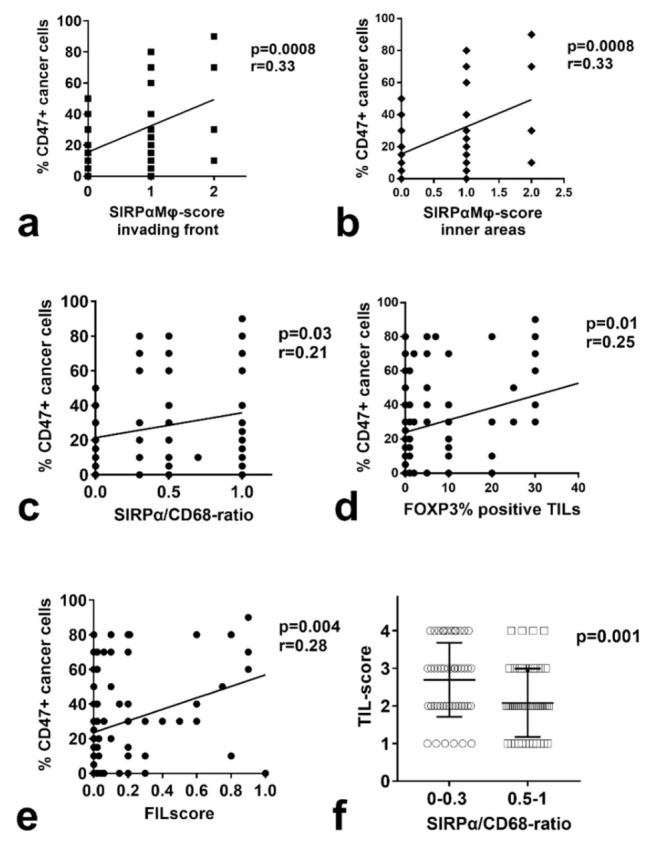
Association between CD47, SIRPα, CD68, and immunological parameters: (**a**,**b**) linear regression analysis between CD47 expression by cancer cells and SIRPα-Mφ score in the invading tumor front and inner tumor areas; (**c**) linear regression analysis between CD47 expression by cancer cells and SIRPα/CD68-ratio; (**d**) linear regression analysis between CD47 expression by cancer cells and the percentage of FOXP3+ TILs; (**e**) linear regression analysis between CD47 expression by cancer cells and FIL-score; (**f**) TIL-score distribution in two SIRPα/CD68-ratio groups.

**Table 1 cancers-14-01801-t001:** Distribution of the score of infiltration of tumor stroma by SIRPα and CD68 expressing macrophages (invading front vs. inner tumor areas). Numbers in brackets aside from numbers refer to the percentage of scored cases in the total of 98 patients.

SIRPα+Mφ (Invading Front)					
SIRPα+Mφ (Inner Areas)	0	1	2	3	Total
0	26	4	0	0	30
	(26/53)	−4.08	0	0	−30.6
1	4	53	5	0	62
	−4.08	−54.08	−5.1	0	−63.2
2	0	1	5	0	6
	0	−1.02	−5.1	0	−6.2
3	0	0	0	0	0
	0	0	0	0	0
Total	30	58	10	0	98
	−30.6	−59.2	−10.2	0	−100
**CD68Mφ** **(Invading Front)**					
**CD68+Mφ** **(Inner Areas)**	**0**	**1**	**2**	**3**	**Total**
0	0	0	0	0	0
	0	0	0	0	0
1	0	28	10	2	40
	0	−28.57	−10.2	−2.04	−40.8
2	0	5	16	7	28
	0	−5.1	−16.37	−7.14	−28.6
3	0	2	2	26	30
	0	−2.04	−2.04	−26.53	−30.6
Total	0	35	28	35	98
	0	−35.7	−28.6	−35.7	−100

**Table 2 cancers-14-01801-t002:** Distribution of SIRPα/CD68-ratio among cases (invading front vs. inner tumor areas). Numbers in brackets aside numbers refer to the percentage of scored cases in the total of 98 patients.

Invading Front						
(Median 0.4)						
Inner Areas (Median 0.5)	0	0.3	0.5	0.7	1	Total
0	25	2	1	0	2	30
	−25.52	−2.04	−1.02	0	−2.04	−30.6
0.3	2	12	2	3	0	19
	−2.04	−12.24	−1.04	−3.06	0	−19.4
0.5	1	4	7	1	2	15
	−1.02	−4.08	−7.14	−1.02	−2.04	−15.3
0.7	0	0	0	1	0	1
	0	0	0	−1.02	(0	−1.02
1	1	1	8	0	23	33
	−1.02	−1.02	−18.4	0	−23.46	−33.7
Total	29	19	18	5	27	98
	−29.6	−19.4	−18.4	−5.1	−27.5	−100

## Data Availability

Data are archieved in our Pathology Department files, and can become available upon reasonable request.

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
