# Peer review of "Expression of CD47 and SIRPα Macrophage Immune-Checkpoint Pathway in Non-Small-Cell Lung Cancer"

_cancers, 2022, doi:10.3390/cancers14071801_

Round 1

Reviewer 1 Report

accept

Reviewer 2 Report

accept

This manuscript is a resubmission of an earlier submission. The following is a list of the peer review reports and author responses from that submission.

Round 1

Reviewer 1 Report

This manuscript presents a retrospective immunohistochemical analysis of CD47, SIRPα and CD68 protein expression in tissue sections from 98 cases of NSCLC. Given that the three cited previous studies reached contradictory conclusions regarding CD47 in NSCLC, it is unclear whether small numbers of three NSCLC histologies should be lumped together for this study. The case numbers of each histology are insufficient to assess whether CD47 and SIRPα have differential functions in adenocarcinomas, squamous carcinomas, and undifferentiated carcinomas, which are masked when these divergent cancers are studied in aggregate. The authors’ presumption that correlations between protein expression relates to the known antiphagocytic function of CD47 interacting with SIRPα further detracts from the study as presented.

  1. The manuscript is presented primarily as a study of CD47 in NSCLC, but the strongest data is for two macrophage markers, SIRPα and CD68. It is unclear whether the significance of SIRPα relates in any way with its immune checkpoint function as a CD47 counter-receptor as opposed to simply being another macrophage marker like CD68. Is SIRPα significant independent of CD68?
  2. Although the Simple Summary states “High expression, however, of SIRPα by CD68+ TAMs was linked with CD47 expression by cancer cells”. This overstates the data which merely correlates SIRPα with CD47 in a minor subset of inner tumor cells.
  3. Similarly, the Conclusions state “Activation of phagocytosis inhibition pathways through CD47/SIRPα up-regulation occurs in a third about of NSCLCs, is linked with poor prognosis, and is a critical immune checkpoint target for the development of adjuvant immunotherapy policies, aiming to improve the cure rates in operable NSCLC.” However, no data is presented establishing CD47/SIRPα as a functional antiphagocytic  immune checkpoint in NSCLC. This unfounded speculation should be deleted.
  4. Therapeutic implications of the data are overstated. For example, given that “one third expressed CD47 in more than 50% of cancer cells”, the converse is that two thirds express CD47 in less than 50% of the cancer cells. Therefore, a CD47 therapeutic antibody would be expected to not induce clearance of most cancer cells in most NSCLC cases, and the negative cancer cells would probably mediate resistant regrowth. Thus, it is difficult to understand why CD47 antibodies would be expected to have therapeutic benefit for NSCLC as proposed.
  5. The Introduction cites only a small fraction of the published studies relating CD47 expression with clinical behavior and prognosis of cancers, and studies that contradict some of the cited studies are omitted.
  6. This statement is not true: “Regarding SIRPα expression, its clinicopathological and prognostic value in solid tumors remains unknown.” For example PMID: 34009732, PMID: 33799989, PMID: 28097229, PMID: 12773380

Reviewer 2 Report

The manuscript submitted by Koukourakis, et al. investigated the expression of CD47 and SIRPα in NSCL cancer. Interactions between CD47 and SIRPa affect the proliferation, migration, invasion, and apoptosis of cancer cells, as well as the activation of immune cells. Therefore the subject of this study is very meaningful. The authors testified the extensive expression of membranous CD47  in 1/3 of the tested cancer cells, and the expression of SIRPα  by tumor-associated macrophages. However, these and most of other results are within the expectation of experts or known facts, or at best the known facts are reconfirmed in NSCL cancer. Thus the implication and significance of these findings are limited.  Moreover, the experimental data are not sufficient and the methodology is simplex. Other supportive data like protein expression and immunofluorescence images are required. So, my conclusion is: the paper cannot be accepted in the current status.